# Causal Confusion in Imitation Learning

**Pim de Haan**[*1]**, Dinesh Jayaraman**[†‡]**, Sergey Levine**[†]
[*]Qualcomm AI Research, University of Amsterdam,
[†]Berkeley AI Research, [‡] Facebook AI Research

## Abstract

Behavioral cloning reduces policy learning to supervised learning by training a discriminative model to predict expert actions given observations. Such discriminative models are non-causal: the training procedure is unaware of the causal structure of the interaction between the expert and the environment. We point out that ignoring causality is particularly damaging because of the distributional shift in imitation learning. In particular, it leads to a counter-intuitive "causal misidentification" phenomenon: access to more information can yield worse performance. We investigate how this problem arises, and propose a solution to combat it through targeted interventions—either environment interaction or expert queries—to determine the correct causal model. We show that causal misidentification occurs in several benchmark control domains as well as realistic driving settings, and validate our solution against DAgger and other baselines and ablations.

## 1 Introduction

Imitation learning allows for control policies to be learned directly from example demonstrations provided by human experts. It is easy to implement, and reduces or removes the need for extensive interaction with the environment during training [58, 41, 4, 1, 20].

However, imitation learning suffers from a fundamental problem: distributional shift [9, 42]. Training and testing state distributions are different, induced respectively by the expert and learned policies. Therefore, imitating expert actions on expert trajectories may not align with the true task objective. While this problem is widely acknowledged [41, 9, 42, 43], yet with careful engineering, naïve behavioral cloning approaches have yielded good results for several practical problems [58, 41, 44, 36, 37, 4, 33, 3]. This raises the question: is distributional shift really still a problem?

In this paper, we identify a somewhat surprising and very problematic effect of distributional shift: "causal misidentification." Distinguishing correlates of expert actions in the demonstration set from true causes is usually very difficult, but may be ignored without adverse effects when training and testing distributions are identical (as assumed in supervised learning), since nuisance correlates continue to hold in the test set. However, this can cause catastrophic problems in imitation learning due to distributional shift. This is exacerbated by the causal structure of sequential action: the very fact that current actions cause future observations often introduces complex new nuisance correlates.

To illustrate, consider behavioral cloning to train a neural network to drive a car. In scenario A, the model's input is an image of the dashboard and windshield, and in scenario B, the input to the model (with identical architecture) is the same image but with the dashboard masked out (see Fig 1). Both cloned policies achieve low training loss, but when tested on the road, model B drives well, while model A does not. The reason: the dashboard has an indicator light that comes on immediately when the brake is applied, and model A wrongly learns to apply the brake only when the brake light is on. Even though the brake light is the *effect* of braking, model A could achieve low training error by *misidentifying* it as the cause instead.

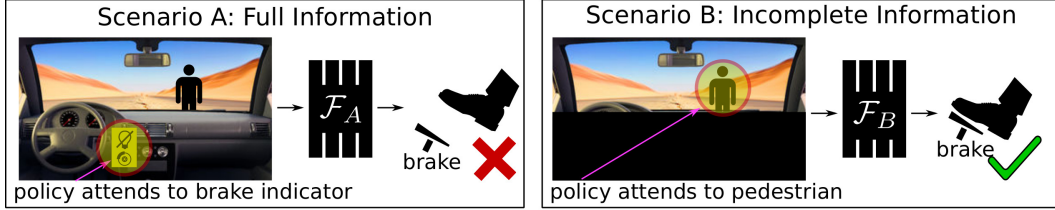

Figure 1: Causal misidentification: *more* information yields worse imitation learning performance. Model A relies on the braking indicator to decide whether to brake. Model B instead correctly attends to the pedestrian.

This situation presents a give-away symptom of causal misidentification: access to *more information* leads to *worse generalization performance* in the presence of distributional shift. Causal misidentification occurs commonly in natural imitation learning settings, especially when the imitator's inputs include history information.

In this paper, we first point out and investigate the causal misidentification problem in imitation learning. Then, we propose a solution to overcome it by learning the correct causal model, even when using complex deep neural network policies. We learn a mapping from causal graphs to policies, and then use targeted interventions to efficiently search for the correct policy, either by querying an expert, or by executing selected policies in the environment.

## 2 Related Work

**Imitation learning.** Imitation learning through behavioral cloning dates back to Widrow and Smith, 1964 [58], and has remained popular through today [41, 44, 36, 37, 4, 13, 33, 56, 3]. The distributional shift problem, wherein a cloned policy encounters unfamiliar states during autonomous execution, has been identified as an issue in imitation learning [41, 9, 42, 43, 25, 19, 3]. This is closely tied to the "feedback" problem in general machine learning systems that have direct or indirect access to their own past states [47, 2]. For imitation learning, various solutions to this problem have been proposed [9, 42, 43] that rely on iteratively querying an expert based on states encountered by some intermediate cloned policy, to overcome distributional shift; DAgger [43] has come to be the most widely used of these solutions.

We show evidence that the distributional shift problem in imitation learning is often due to causal misidentification, as illustrated schematically in Fig 1. We propose to address this through targeted interventions on the states to learn the true causal model to overcome distributional shift. As we will show, these interventions can take the form of either environmental rewards with no additional expert involvement, or of expert queries in cases where the expert is available for additional inputs. In expert query mode, our approach may be directly compared to DAgger [43]: indeed, we show that we successfully resolve causal misidentification using orders of magnitude fewer queries than DAgger.

We also compare against Bansal et al. [3]: to prevent imitators from copying past actions, they train with dropout [53] on dimensions that might reveal past actions. While our approach seeks to find the true causal graph in a mixture of graph-parameterized policies, dropout corresponds to directly applying the mixture policy. In our experiments, our approach performs significantly better.

**Causal inference.** Causal inference is the general problem of deducing cause-effect relationships among variables [52, 38, 40, 50, 10, 51]. "Causal discovery" approaches allow causal inference from pre-recorded observations under constraints [54, 17, 29, 15, 30, 31, 26, 14, 34, 57]. Observational causal inference is known to be impossible in general [38, 39]. We operate in the interventional regime [55, 11, 49, 48] where a user may "experiment" to discover causal structures by assigning values to some subset of the variables of interest and observing the effects on the rest of the system. We propose a new interventional causal inference approach suited to imitation learning. While ignoring causal structure is particularly problematic in imitation learning, ours is the first effort directly addressing this, to our knowledge.

## 3 The Phenomenon of Causal Misidentification

In imitation learning, an expert demonstrates how to perform a task (e.g., driving a car) for the benefit of an agent. In each demo, the agent has access both to its $n$-dim. state observations at each time $t$, $X^t = [X_1^t, X_2^t, \dots X_n^t]$ (e.g., a video feed from a camera), and to the expert's action $A^t$ (e.g., steering,

acceleration, braking). Behavioral cloning approaches learn a mapping $\pi$ from $X^t$ to $A^t$ using all $(X^t, A^t)$ tuples from the demonstrations. At test time, the agent observes $X^t$ and executes $\pi(X^t)$.

The underlying sequential decision process has complex causal structures, represented in Fig 2. States influence future expert actions, and are also themselves influenced by past actions and states.

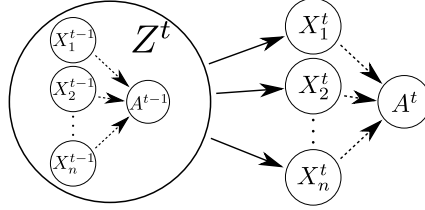

In particular, expert actions $A^t$ are influenced by *some* information in state $X^t$, and unaffected by the rest. For the moment, assume that the dimensions $X_1^t, X_2^t, X_3^t, \ldots$ of $X^t$ represent disentangled factors of variation. Then some unknown subset of these factors ("causes") affect expert actions, and the rest do not ("nuisance variables").

Figure 2: Causal dynamics of imitation. Parents of a node represent its causes.

A confounder $Z^t = [X^{t-1}, A^{t-1}]$ influences each state variable in $X^t$, so that some nuisance variables may still be correlated with $A^t$ among $(X^t, A^t)$ pairs from demonstrations. In Fig 1, the dashboard light is a nuisance variable.

A naïve behavioral cloned policy might rely on nuisance correlates to select actions, producing low training error, and even generalizing to held-out $(X^t, A^t)$ pairs. However, this policy must contend with distributional shift when deployed: actions $A_t$ are chosen by the *imitator* rather than the expert, affecting the distribution of $Z^t$ and $X^t$. This in turn affects the policy mapping from $X^t$ to $A^t$, leading to poor performance of expert-cloned policies. We define "causal misidentification" as the phenomenon whereby cloned policies fail by misidentifying the causes of expert actions.

## 3.1 Robustness and Causality in Imitation Learning

Intuitively, distributional shift affects the relationship of the expert action $A^t$ to nuisance variables, but not to the true causes. In other words, to be maximally robust to distributional shift, a policy must rely solely on the true causes of expert actions, thereby avoiding causal misidentification. This intuition can be formalized in the language of functional causal models (FCM) and interventions [38].

**Functional causal models:** A functional causal model (FCM) over a set of variables $\{Y_i\}_{i=1}^n$ is a tuple $(G, \theta_G)$ containing a graph $G$ over $\{Y_i\}_{i=1}^n$, and deterministic functions $f_i(\cdot; \theta_G)$ with parameters $\theta_G$ describing how the causes of each variable $Y_i$ determine it: $Y_i = f_i(Y_{\text{Pa}(i;G)}, E_i; \theta_G)$, where $E_i$ is a stochastic noise variable that represents all external influences on $Y_i$, and $\text{Pa}(i; G)$ denote the indices of parent nodes of $Y_i$, which correspond to its causes.

An "intervention" $do(Y_i)$ on $Y_i$ to set its value may now be represented by a structural change in this graph to produce the "mutilated graph" $G_{\bar{Y}_i}$, in which incoming edges to $Y_i$ are removed.[1]

Applying this formalism to our imitation learning setting, any distributional shift in the state $X^t$ may be modeled by intervening on $X^t$, so that correctly modeling the "interventional query" $p(A^t|do(X^t))$ is sufficient for robustness to distributional shifts. Now, we may formalize the intuition that only a policy relying solely on true causes can robustly model the mapping from states to optimal/expert actions under distributional shift.

In Appendix B, we prove that under mild assumptions, correctly modeling interventional queries does indeed require learning the correct causal graph $G$. In the car example, "setting" the brake light to on or off and observing the expert's actions would yield a clear signal unobstructed by confounders: the brake light does not affect the expert's braking behavior.

## 3.2 Causal Misidentification in Policy Learning Benchmarks and Realistic Settings

Before discussing our solution, we first present several testbeds and real-world cases where causal misidentification adversely influences imitation learning performance.

**Control Benchmarks.** We show that causal misidentification is induced with small changes to widely studied benchmark control tasks, simply by adding more information to the state, which intuitively ought to make the tasks easier, not harder. In particular, we add information about the previous action, which tends to correlate with the current action in the expert data for many standard control problems. This is a proxy for scenarios like our car example, in which correlates of past actions

are observable in the state, and is similar to what we might see from other sources of knowledge about the past, such as memory or recurrence. We study three kinds of tasks: (i) MountainCar (continuous states, discrete actions), (ii) MuJoCo Hopper (continuous states and actions), (iii) Atari games: Pong, Enduro and UpNDown (states: two stacked consecutive frames, discrete actions).

For each task, we study imitation learning in two scenarios. In scenario A (henceforth called "CONFOUNDED"), the policy sees the augmented observation vector, including the previous action. In the case of low-dimensional observations, the state vector is expanded to include the previous action at an index that is unknown to the learner. In the case of image observations, we overlay a symbol corresponding to the previous action at an unknown location on the image (see Fig 3). In scenario B

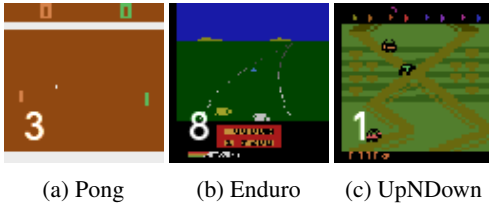

(a) Pong      (b) Enduro      (c) UpNDown

Figure 3: The Atari environments with indicator of past action (white number in lower left).

("ORIGINAL"), the previous action variable is replaced with random noise for low-dimensional observations. For image observations, the original images are left unchanged. Demonstrations are generated synthetically as described in Appendix A. In all cases, we use neural networks with identical architectures to represent the policies, and we train them on the same demonstrations.

Fig 4 shows the rewards against varying demonstration dataset sizes for MountainCar, Hopper, and Pong. Appendix E shows additional results, including for Enduro and UpNDown. All policies are trained to near-zero validation error on held-out expert state-action tuples. ORIGINAL produces rewards tending towards expert performance as the size of the imitation dataset increases. CONFOUNDED either requires many more demonstrations to reach equivalent performance, or fails completely to do so.

Overall, the results are clear: across these tasks, access to *more* information leads to inferior performance. As Fig 11 in the appendix shows, this difference is not due to different training/validation losses on the expert demonstrations—for example, in Pong, CONFOUNDED produces lower validation loss than ORIGINAL on held-out demonstration samples, but produces lower rewards when actually used for control. These results not only validate the existence of causal misidentification, but also provides us with testbeds for investigating a potential solution.

**Real-World Driving.** Our testbeds introduce deliberate nuisance variables to the "original" observation variables for ease of evaluation, but evidence suggests that misattribution is pervasive in common real-world imitation learning settings. Real-world problems often have no privileged "original" observation space, and very natural-seeming state spaces may still include nuisance factors— as in our dashboard light setting (Fig 1), where causal misattribution occurs when using the full image from the camera.

In particular, history would seem a natural part of the state space for real-world driving, yet recurrent/history-based imitation has been consistently observed in prior work to hurt performance, thus exhibiting clear symptoms of causal misidentification [36, 56, 3]. While these histories contain valuable information for driving, they also naturally introduce information about nuisance factors such as previous actions. In all three cases, more information led to worse results for the behavioral cloning policy, but this was neither attributed specifically to causal misidentification, nor tackled using causally motivated approaches.

We draw the reader's attention to particularly telling results from Wang et al. [56] for learning to drive in near-photorealistic GTA-V [24] environments, using behavior cloning with DAgger-inspired expert perturbation. Imitation learning policies are trained using overhead image observations with and without "history" information (HISTORY and NO-HISTORY) about the ego-position trajectory of the car in the past.

| Metrics → Methods ↓ | Validation Perplexity | Driving Performance | | |
|---|---|---|---|---|
| | | Distance | Interventions | Collisions |
| HISTORY | **0.834** | 144.92 | 2.94 ± 1.79 | 6.49 ± 5.72 |
| NO-HISTORY | 0.989 | **268.95** | **1.30 ± 0.78** | **3.38 ± 2.55** |

Table 1: Imitation learning results from Wang et al. [56]. Accessing history yields better validation performance, but worse actual driving performance.

Similar to our tests above, architectures are identical for the two methods. And once again, like in our tests above, HISTORY has better performance on held-out demonstration data, but much worse performance when actually deployed. Tab 1 shows these results, reproduced from Wang et al. [56] Table II. These results constitute strong evidence for the prevalence of causal misidentification in

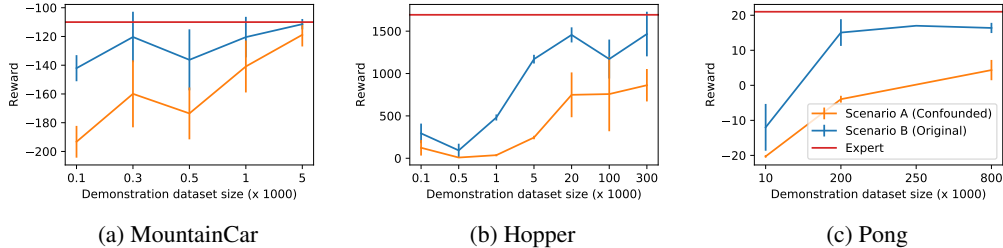

| (a) MountainCar | (b) Hopper | (c) Pong |
|---|---|---|

Figure 4: Diagnosing causal misidentification: net reward (y-axis) vs number of training samples (x-axis) for ORIGINAL and CONFOUNDED, compared to expert reward (mean and stdev over 5 runs). Also see Appendix E.

realistic imitation learning settings. Bansal et al. [3] also observe similar symptoms in a driving setting, and present a dropout [53] approach to tackle it, which we compare to in our experiments. Subsequent to an earlier version of this work, Codevilla et al. [8] also verify causal confusion in realistic driving settings, and propose measures to address a specific instance of causal confusion.

# 4 Resolving Causal Misidentification

Recall from Sec 3.1 that robustness to causal misidentification can be achieved by finding the true causal model of the expert's actions. We propose a simple pipeline to do this. First, we jointly learn policies corresponding to various causal graphs (Sec 4.1). Then, we perform targeted interventions to efficiently search over the hypothesis set for the correct causal model (Sec 4.2).

## 4.1 Causal Graph-Parameterized Policy Learning

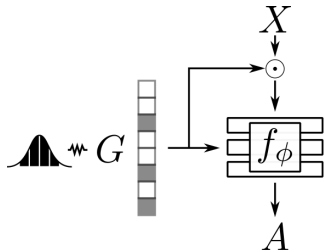

Figure 5: Graph-parameterized policy.

In this step, we learn a policy corresponding to each candidate causal graph. Recall from Sec 3 that the expert's actions $A$ are based on an unknown subset of the state variables $\{X_i\}_{i=1}^n$. Each $X_i$ may either be a cause or not, so there are $2^n$ possible graphs. We parameterize the structure $G$ of the causal graph as a vector of $n$ binary variables, each indicating the presence of an arrow from $X_k$ to $A$ in Fig 2. We then train a single graph-parameterized policy $\pi_G(X) = f_\phi([X \odot G, G])$, where $\odot$ is element-wise multiplication, and $[\cdot, \cdot]$ denotes concatenation. $\phi$ are neural network parameters, trained through gradient descent to minimize:

$$\mathbb{E}_G[\ell(f_\phi([X_i \odot G, G]), A_i)], \tag{1}$$

where $G$ is drawn uniformly at random over all $2^n$ graphs and $\ell$ is a mean squared error loss for the continuous action environments and a cross-entropy loss for the discrete action environments. Fig 5 shows a schematic of the training time architecture. The policy network $f_\phi$ mapping observations $X$ to actions $A$ represents a mixture of policies, one corresponding to each value of the binary causal graph structure variable $G$, which is sampled as a bernoulli random vector.

In Appendix D, we propose an approach to perform variational Bayesian causal discovery over graphs $G$, using a latent variable model to infer a distribution over functional causal models (graphs and associated parameters)—the modes of this distribution are the FCMs most consistent with the demonstration data. This resembles the scheme above, except that instead of uniform sampling, graphs are sampled preferentially from FCMs that fit the training demonstrations well. We compare both approaches in Sec 5, finding that simple uniform sampling nearly always suffices in preparation for the next step: targeted intervention.

## 4.2 Targeted Intervention

Having learned the graph-parameterized policy as in Sec 4.1, we propose targeted intervention to compute the likelihood $\mathcal{L}(G)$ of each causal graph structure hypothesis $G$. In a sense, imitation learning provides an ideal setting for studying interventional causal learning: causal misidentification presents a clear challenge, while the fact that the problem is situated in a sequential decision process where the agent can interact with the world provides a natural mechanism for carrying out limited interventions.

We propose two intervention modes, both of which can be carried out by interaction with the environment via the actions:

**Expert query mode.** This is the standard intervention approach applied to imitation learning: intervene on $X^t$ to assign it a value, and observe the expert response $A$. To do this, we sample a graph $G$ at the beginning of each intervention episode and execute the policy $\pi_G$. Once data is collected in this manner, we elicit expert labels on interesting states. This requires an interactive expert, as in DAgger [42], but requires substantially fewer expert queries than DAgger, because: (i) the queries serve only to disambiguate among a relatively small set of valid FCMs, and (ii) we use disagreement among the mixture of policies in $f_\phi$ to query the expert efficiently in an active learning approach. We summarize this approach in Algorithm 1.

**Policy execution mode.** It is not always possible to query an expert. For example, for a learner learning to drive a car by watching a human driver, it may not be possible to put the human driver into dangerous scenarios that the learner might encounter at intermediate stages of training. In cases like these where we would like to learn from pre-recorded demonstrations alone, we propose to intervene indirectly by using environmental returns (sum of rewards over time in an episode) $R = \sum_t r_t$. The policies $\pi_G(\cdot) = f_\phi([\cdot \odot G, G])$ corresponding to different hypotheses $G$ are executed in the

---

**Algorithm 1** Expert query intervention

**Input:** policy network $f_\phi$ s.t. $\pi_G(X) = f_\phi([X \odot G, G])$
Initialize $w = 0, \mathcal{D} = \emptyset$.
Collect states $\mathcal{S}$ by executing $\pi_{mix}$, the mixture of policies $\pi_G$ for uniform samples $G$.
For each $X$ in $S$, compute disagreement score:
$\quad D(X) = \mathbb{E}_G[D_{KL}(\pi_G(X), \pi_{mix}(X))]$
Select $\mathcal{S}' \subset \mathcal{S}$ with maximal $D(X)$.
Collect state-action pairs $\mathcal{T}$ by querying expert on $\mathcal{S}'$.
**for** $i = 1 \ldots N$ **do**
$\quad$ Sample $G \sim p(G) \propto \exp\langle w, G \rangle$.
$\quad \mathcal{L} \leftarrow \mathbb{E}_{s,a \sim \mathcal{T}}[\ell(\pi_G(s), a)]$
$\quad \mathcal{D} \leftarrow \mathcal{D} \cup \{(G, \mathcal{L})\}$
$\quad$ Fit $w$ on $\mathcal{D}$ with linear regression.
**end for**
**Return:** $\arg\max_G p(G)$

---

**Algorithm 2** Policy execution intervention

**Input:** policy network $f_\phi$ s.t. $\pi_G(X) = f_\phi([X \odot G, G])$
Initialize $w = 0, \mathcal{D} = \emptyset$.
**for** $i = 1 \ldots N$ **do**
$\quad$ Sample $G \sim p(G) \propto \exp\langle w, G \rangle$.
$\quad$ Collect episode return $R_G$ by executing $\pi_G$.
$\quad \mathcal{D} \leftarrow \mathcal{D} \cup \{(G, R_G)\}$
$\quad$ Fit $w$ on $\mathcal{D}$ with linear regression.
**end for**
**Return:** $\arg\max_G p(G)$

---

environment and the returns $R_G$ collected. The likelihood of each graph is proportional to the exponentiated returns $\exp R_G$. The intuition is simple: environmental returns contain information about optimal expert policies even when experts are not queryable. Note that we do not even assume access to per-timestep rewards as in standard reinforcement learning; just the *sum* of rewards for each completed run. As such, this intervention mode is much more flexible. See Algorithm 2.

Note that both of the above intervention approaches evaluate individual hypotheses in isolation, but the number of hypotheses grows exponentially in the number of state variables. To handle larger states, we infer a graph distribution $p(G)$, by assuming an energy based model with a linear energy $E(G) = \langle w, G \rangle + b$, so the graph distribution is $p(G) = \prod_i p(G_i) = \prod_i \text{Bernoulli}(G_i | \sigma(w_i/\tau))$, where $\sigma$ is the sigmoid, which factorizes in independent factors. The independence assumption is sensible as our approach collapses $p(G)$ to its mode before returning it and the collapsed distribution is always independent. $E(G)$ is inferred from linear regression on the likelihoods. This process is depicted in Algorithms 1 and 2. The above method can be formalized within the reinforcement learning framework [27]. As we show in Appendix H, the energy-based model can be seen as an instance of soft Q-learning [16].

## 4.3 Disentangling Observations

In the above, we have assumed access to disentangled observations $X^t$. When this is not the case, such as with image observations, $X^t$ must be set to a disentangled representation of the observation at time $t$. We construct such a representation by training a $\beta$-VAE [22, 18] to reconstruct the original observations. To capture states beyond those encountered by the expert, we train with a mix of expert and random trajectory states. Once trained, $X^t$ is set to be the mean of the latent distribution produced at the output of the encoder. The VAE training objective encourages disentangled dimensions in the latent space [5, 6]. We employ CoordConv [28] in both the encoder and the decoder architectures.

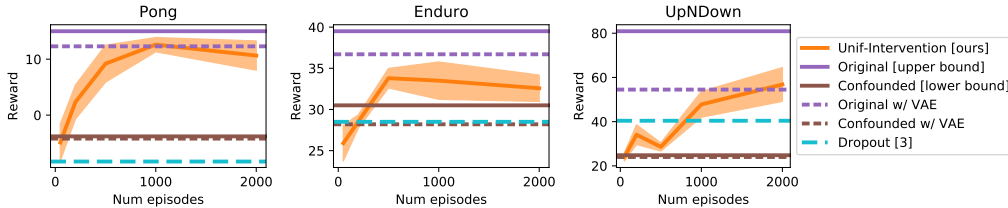

Figure 6: Reward vs. number of intervention episodes (policy execution interventions) on Atari games. UNIF-INTERVENTION succeeds in getting rewards close to ORIGINAL W/ VAE, while the DROPOUT baseline only outperforms CONFOUNDED W/ VAE in UpNDown.

## 5    Experiments

We now evaluate the solution described in Sec 4 on the five tasks (MountainCar, Hopper, and 3 Atari games) described in Sec 3.2. In particular, recall that CONFOUNDED performed significantly worse than ORIGINAL across all tasks. In our experiments, we seek to answer the following questions: **(1)** Does our targeted intervention-based solution to causal misidentification bridge the gap between CONFOUNDED and ORIGINAL? **(2)** How quickly does performance improve with intervention? **(3)** Do both intervention modes (expert query, policy execution) described in Sec 4.2 resolve causal misidentification? **(4)** Does our approach in fact recover the true causal graph? **(5)** Are disentangled state representations necessary?

In each of the two intervention modes, we compare two variants of our method: UNIF-INTERVENTION and DISC-INTERVENTION. They only differ in the training of the graph-parameterized mixture-of-policies $f_\phi$—while UNIF-INTERVENTION samples causal graphs uniformly, DISC-INTERVENTION uses the variational causal discovery approach mentioned in Sec 4.1, and described in detail in Appendix D.

**Baselines.**   We compare our method against three baselines applied to the confounded state. DROPOUT trains the policy using Eq 1 and evaluates with the graph $G$ containing all ones, which amounts to dropout regularization [53] during training, as proposed by Bansal et al. [3]. DAGGER [42] addresses distributional shift by querying the expert on states encountered by the imitator, requiring an interactive expert. We compare DAGGER to our expert query intervention approach. Lastly, we compare to Generative Adversarial Imitation Learning (GAIL) [19]. GAIL is an alternative to standard behavioral cloning that works by matching demonstration trajectories to those generated by the imitator during roll-outs in the environment. Note that the PC algorithm [26], commonly used in causal discovery from passive observational data, relies on the faithfulness assumption, which causes it to be infeasible in our setting, as explained in Appendix C. See Appendices B & D for details.

**Intervention by policy execution.**   Fig 7 plots episode rewards versus number of policy execution intervention episodes for MountainCar and Hopper. The reward always corresponds to the current mode $\arg\max_G p(G)$ of the posterior distribution over graphs, updated after each episode, as described in Algorithm 2. In these cases, both UNIF-INTERVENTION and DISC-INTERVENTION eventually converge to models yielding similar rewards, which we verified to be the correct causal model i.e., true causes are selected and nuisance correlates left out. In early episodes on MountainCar, DISC-INTERVENTION benefits from the prior over graphs inferred in the variational causal discovery phase. However, in Hopper, the simpler UNIF-INTERVENTION performs just as well. DROPOUT does indeed help in both settings, as reported in Bansal et al. [3], but is significantly poorer than our

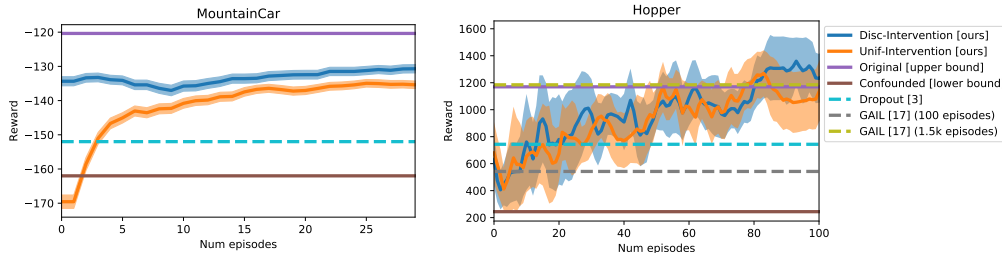

Figure 7: Reward vs. number of intervention episodes (policy execution interventions) on MountainCar and Hopper. Our methods UNIF-INTERVENTION and DISC-INTERVENTION bridge most of the causal misidentification gap (between ORIGINAL (lower bound) and CONFOUNDED (upper bound), approaching ORIGINAL performance after tens of episodes. GAIL [19] (on Hopper) achieves this too, but after 1.5k episodes.

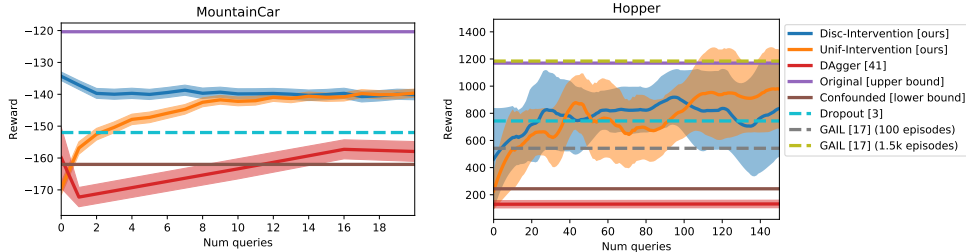

Figure 8: Reward vs. expert queries (expert query interventions) on MountainCar and Hopper. Our methods partially bridge the gap from CONFOUNDED (lower bd) to ORIGINAL (upper bd), also outperforming DAGGER [43] and DROPOUT [3]. GAIL [19] outperforms our methods on Hopper, but requires a large number of policy roll-outs (also see Fig 7 comparing GAIL to our policy execution-based approach).

approach variants. GAIL requires about 1.5k episodes on Hopper to match the performance of our approaches, which only need tens of episodes. Appendix G further analyzes the performance of GAIL. Standard implementations of GAIL do not handle discrete action spaces, so we do not evaluate it on MountainCar.

As described in Sec 4.3, we use a VAE to disentangle image states in Atari games to produce 30-D representations for Pong and Enduro and 50-D representations for UpNDown. We set this dimensionality heuristically to be as small as possible, while still producing good reconstructions as assessed visually. Requiring the policy to utilize the VAE representation without end-to-end training does result in some drop in performance, as seen in Fig 6. However, causal misidentification still causes a very large drop of performance even relative to the baseline VAE performance. DISC-INTERVENTION is hard to train as the cardinality of the state increases, and yields only minor advantages on Hopper (14-D states), so we omit it for these Atari experiments. As Fig 6 shows, UNIF-INTERVENTION indeed improves significantly over CONFOUNDED W/ VAE in all three cases, matching ORIGINAL W/ VAE on Pong and UpNDown, while the DROPOUT baseline only improves UpNDown. In our experiments thus far, GAIL fails to converge to above-chance performance on any of the Atari environments. These results show that our method successfully alleviates causal misidentification within relatively few trials.

**Intervention by expert queries.** Next, we perform direct intervention by querying the expert on samples from trajectories produced by the different causal graphs. In this setting, we can also directly compare to DAGGER [43]. Fig 8 shows results on MountainCar and Hopper. Both our approaches successfully improve over CONFOUNDED within a small number of queries. Consistent with policy execution intervention results reported above, we verify that our approach again identifies the true causal model correctly in both tasks, and also performs better than DROPOUT in both settings. It also exceeds the rewards achieved by DAGGER, while using far fewer expert queries. In Appendix F, we show that DAGGER requires hundreds of queries to achieve similar rewards for MountainCar and tens of thousands for Hopper. Finally, GAIL with 1.5k episodes outperforms our expert query interventions approach. Recall however from Fig 8 that this is an order of magnitude more than the number of episodes required by our policy intervention approach.

Once again, DISC-INTERVENTION only helps in early interventions on MountainCar, and not at all on Hopper. Thus, our method's performance is primarily attributable to the targeted intervention stage, and the exact choice of approach used to learn the mixture of policies is relatively insignificant.

Overall, of the two intervention approaches, policy execution converges to better final rewards. Indeed, for the Atari environments, we observed that expert query interventions proved ineffective. We believe this is because expert agreement is an imperfect proxy for true environmental rewards.

**Interpreting the learned causal graph.** Our method labels each dimension of the VAE encoding of the frame as a cause or nuisance variable. In Fig 9, we analyze these inferences in the Pong environment as follows: in the top row, a frame is encoded into the VAE latent, then for all nuisance dimensions (as inferred by our approach UNIF-INTERVENTION), that dimension is replaced with a sample from the prior, and new samples are generated. In the bottom row, the same procedure is applied with a random graph that has as many nuisance variables as the inferred graph. We observe that in the top row, the causal variables (the ball and paddles) are shared between the samples, while the nuisance variables (the digit) differ, being replaced either with random digits or unreadable digits.

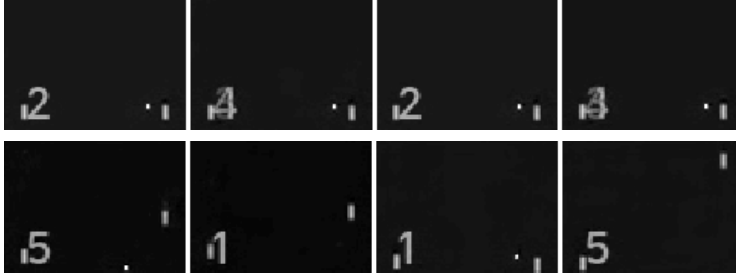

Figure 9: Samples from (top row) learned causal graph and (bottom row) random causal graph. (See text)

In the bottom row, the causal variables differ strongly, indicating that important aspects of the state are judged as nuisance variables. This validates that, consistent with MountainCar and Hopper, our approach does indeed identify true causes in Pong.

**Necessity of disentanglement.** Our intervention method assumes a disentangled representation of state. Otherwise, each of the $n$ individual dimensions in the state might capture both causes as well as nuisance variables and problem of discovering true causes is no longer reducible to searching over $2^n$ graphs.

| Mode | Representation | Reward |
|------|---------------|--------|
| Policy execution | Disentangled | **-137** |
| | Entangled | -145 |
| Expert queries | Disentangled | **-140** |
| | Entangled | -165 |

Table 2: Intervention on (dis)entangled MountainCar.

To test this empirically, we create a variant of our MountainCar CONFOUNDED testbed, where the 3-D past action-augmented state vector is rotated by a fixed, random rotation. After training the graph-conditioned policies on the entangled and disentangled CONFOUNDED state, and applying 30 episodes of policy execution intervention or 20 expert queries, we get the results shown in Tab 2. The results are significantly lower in the entangled than in the disentangled (non-rotated) setting, indicating disentanglement is important for the effectiveness of our approach.

## 6    Conclusions

We have identified a naturally occurring and fundamental problem in imitation learning, "causal misidentification", and proposed a causally motivated approach for resolving it. While we observe evidence for causal misidentification arising in natural imitation learning settings, we have thus far validated our solution in somewhat simpler synthetic settings intended to mimic them. Extending our solution to work for such realistic scenarios is an exciting direction for future work. Finally, apart from imitation, general machine learning systems deployed in the real world also encounter "feedback" [47, 2], which opens the door to causal misidentification. We hope to address these more general settings in the future.

**Acknowledgments:**   We would like to thank Karthikeyan Shanmugam and Shane Gu for pointers to prior work early in the project, and Yang Gao, Abhishek Gupta, Marvin Zhang, Alyosha Efros, and Roberto Calandra for helpful discussions in various stages of the project. We are also grateful to Drew Bagnell and Katerina Fragkiadaki for helpful feedback on an earlier draft of this paper. This project was supported in part by Berkeley DeepDrive, NVIDIA, and Google.

## Footnotes

[1]Work mostly done while at Berkeley AI Research.

[1]For a more thorough overview of FCMs, see [38].

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
