[Supplementary Material]

# A  Expert Demonstrations

To collect demonstrations, we first train an expert with reinforcement learning. We use DQN [35] for MountainCar, TRPO [45] for Hopper, and PPO [46] for the Atari environments (Pong, UpNDown, Enduro). This expert policy is executed in the environment to collect demonstrations.

# B  Necessity of Correct Causal Model

**Faithfulness:** A causal model is said to be faithful when all conditional independence relationships in the distribution are represented in the graph.

We pick up the notation used in Sec 3.1, but for notational simplicity, we drop the time superscript for $X$, $A$, and $Z$ when we are not reasoning about multiple time-steps.

**Proposition 1.** *Let the expert's functional causal model be $(G^*, \theta_{G^*}^*)$, with causal graph $G^* \in \mathcal{G}$ as in Figure 2 and function parameters $\theta_{G^*}^*$. We assume some faithful learner $(\hat{G}, \theta_{\hat{G}}), \hat{G} \in \mathcal{G}$ that agrees on the interventional query:*

$$\forall X, A : p_{G^*, \theta_{G^*}^*}(A|do(X)) = p_{\hat{G}, \theta_{\hat{G}}}(A|do(X))$$

*Then it must be that $G^* = \hat{G}$.[2]*

*Proof.* For graph $G$, define the index set of state variables that are independent of the action in the mutilated graph $G_{\bar{X}}$:

$$I_G = \{i | X_i \underset{G_{\bar{X}}}{\perp\!\!\!\perp} A\}$$

From the assumption of matching interventional queries and the assumption of faithfulness, it follows that: $I_{G^*} = I_{\hat{G}}$. From the graph, we observe that $I_G = \{i | (X_i \to A) \notin G\}$ and thus $G^* = \hat{G}$.   $\square$

# C  Passive Causal Discovery, Faithfulness and Determinism

In many learning scenarios, much information about the causal model can already be inferred passively from the data. This is the problem of causal discovery. Ideally, it would allow us to perform statistical analysis on the random variables in Fig 2 in the demonstration data to determine whether variable $X_i^t$ is a cause of the next expert action $A^t$ or a nuisance variable.

|  | $I(X_i^t; A^t)$ | $I(X_i^t; A^t|Z^t)$ |
|---|---|---|
| $X_0^t$ (cause) | 0.377 | 0.013 |
| $X_1^t$ (cause) | 0.707 | 0.019 |
| $X_2^t$ (nuisance) | 0.654 | 0.027 |

Table 3: Mutual information in bits of the CON-FOUNDED MountainCar setup.

Causal discovery algorithms, such as the PC algorithm [52] test a series of conditional independence relationships in the observed data and construct the set of possible causal graphs whose conditional independence relationships match the data. It does so by assuming *faithfulness*, meaning the joint probability of random variables contains no more conditional independence relationships than the causal graph. In the particular case of the causal model in Fig 2, it is easy to see that $X_i^t$ is a cause of $A^t$, and thus that the arrow $X_i^t \to A^t$ exists, if and only if $X_i^t \not\!\perp\!\!\!\perp A^t|Z^t$, meaning that $X_i^t$ provides extra information about $A^t$ if $Z^t$ is already known.

We test this procedure empirically by evaluating the mutual information $I(X_i^t; A^t|Z^t)$ for the CON-FOUNDED MountainCar benchmark, using the estimator from Gao et al. [12]. The results in Table 3 show that all state variables are correlated with the expert's action, but that all become mostly independent given the confounder $Z^t$, implying none are causes.

Passive causal discovery failed because the critical faithfulness assumption is violated in the MountainCar case. Whenever a state variable $X_i^t$ is a deterministic function of the past $Z^t$, so that $X_i^t \perp\!\!\!\perp A^t|Z^t$ always holds and a passive discovery algorithm concludes no arrow $X_i^t \to A^t$ exists. Such a deterministic transition function for at least a part of the state is very common in realistic imitation learning scenarios, making passive causal discovery inapplicable. Active interventions must thus be used to determine the causal model.

# D Variational Causal Discovery

Figure 10: Training architecture for variational inference-based causal discovery as described in Appendix D. The policy network $f_\phi$ represents a mixture of policies, one corresponding to each value of the binary causal graph structure variable $G$. This variable in turn is sampled from the distribution $q_\psi(G|u)$ produced by an inference network from an input latent $U$. Further, a network $b_\eta$ regresses back to the latent $U$ to enforce that $G$ should not be independent of $U$.

The problem of discovering causal graphs from passively observed data is called causal discovery. The PC algorithm [52] is arguably the most widely used and easily implementable causal discovery algorithm. In the case of Fig 2, the PC algorithm would imply the absence of the arrow $X_i^t \to A^t$, if the conditional independence relation $A^t \perp\!\!\!\perp X_i^t | Z^t$ holds, which can be tested by measuring the mutual information. However, the PC algorithm relies on *faithfulness* of the causal graph. That is, conditional independence must imply d-separation in the graph. However, faithfulness is easily violated in a Markov decision process. If for some $i$, $X_i^t$ is a cause of the expert's action $A^t$ (the arrow $X_i^t \to A^t$ should exist), but $X_i^t$ is the result of a deterministic function of $Z^t$, then always $A^t \perp\!\!\!\perp X_i^t | Z^t$ and the PC algorithm would wrongly conclude that the arrow $X_i^t \to A^t$ is absent.[3]

We take a Bayesian approach to causal discovery [17] from demonstrations. Recall from Sec 3 that the expert's actions $A$ are based on an unknown subset of the state variables $\{X_i\}_{i=1}^n$. Each $X_i$ may either be a cause or not, so there are $2^n$ possible graphs. We now define a variational inference approach to infer a distribution over functional causal models (graphs and associated parameters) such that its modes are consistent with the demonstration data $D$.

While Bayesian inference is intractable, variational inference can be used to find a distribution that is close to the true posterior distribution over models. We parameterize the structure $G$ of the causal graph as a vector of $n$ correlated Bernoulli random variables $G_k$, each indicating the presence of a causal arrow from $X_k$ to $A$. We assume a variational family with a point estimate $\theta_G$ of the parameters corresponding to graph $G$ and use a latent variable model to describe the correlated Bernoulli variables, with a standard normal distribution $q(U)$ over latent random variable $U$:

$$q_\psi(G, \theta) = q_\psi(G)[\theta = \theta_G]$$

$$= \int q(U) \prod_{k=1}^n q_\psi(G_k|U)[\theta = \theta_G] dU$$

We now optimise the evidence lower bound (ELBO):

$$\arg\min_q D_{KL}(q_\psi(G, \theta)|p(G, \theta|D)) =$$

$$\arg\max_{\psi, \theta} \sum_i \mathbb{E}_{U \sim q(U), G_k \sim q_\psi(G_k|U)} \tag{2}$$

$$[\log \pi(A_i|X_i, G, \theta_G) + \log p(G) + \mathcal{H}_q(G) \tag{3}$$

**Likelihood** $\pi(A_i|X_i, G, \theta_G)$ is the likelihood of the observations $X$ under the FCM $(G, \theta_G)$. It is modelled by a single neural network $f_\phi([X \odot G, G])$, where $\odot$ is the element-wise multiplication, $[\cdot, \cdot]$ denotes concatenation and $\phi$ are neural network parameters.

**Entropy**    The entropy term of the KL divergence, $\mathcal{H}_q$, acts as a regularizer to prevent the graph distribution from collapsing to the maximum a-posteriori estimate. It is intractable to directly maximize entropy, but a tractable variational lower bound can be formulated. Using the product rule of entropies, we may write:

$$\mathcal{H}_q(G) = \mathcal{H}_q(G|U) - \mathcal{H}_q(U|G) + \mathcal{H}_q(U)$$
$$= \mathcal{H}_q(G|U) + I_q(U;G)$$

In this expression, $\mathcal{H}_q(G|U)$ promotes diversity of graphs, while $I_q(U;G)$ encourages correlation among $\{G_k\}$. $I_q(U;G)$ can be bounded below using the same variational bound used in InfoGAN [7], with a variational distribution $b_\eta$: $I_q(U;G) \geq \mathbb{E}_{U,G\sim q_\psi} \log b_\eta(U|G)$. Thus, during optimization, in lieu of entropy, we maximize the following lower bound:

$$\mathcal{H}_q(G) \geq \mathbb{E}_{U,G\sim q} \left[ -\sum_k \log q_\psi(G_k|U) + \log b_\eta(U|G) \right]$$

**Prior**    The prior $p(G)$ over graph structures is set to prefer graphs with fewer causes for action $A$—it is thus a sparsity prior:

$$p(G) \propto \exp \sum_k [G_k = 1]$$

**Optimization**    Note that $G$ is a discrete variable, so we cannot use the reparameterization trick [22]. Instead, we use the Gumbel Softmax trick [21, 32] to compute gradients for training $q_\psi(G_k|U)$. Note that this does not affect $f_\phi$, which can be trained with standard backpropagation.

The loss of Eq 3 is easily interpretable independent of the formalism of variational Bayesian causal discovery. A mixture of predictors $f_\phi$ is jointly trained, each paying attention to diverse sparse subsets (identified by $G$) of the inputs. This is related to variational dropout [23]. Once this model is trained, $q_\psi(G)$ represents the hypothesis distribution over graphs, and $\pi_G(x) = f_\phi([x \odot G, G])$ represents the imitation policy corresponding to a graph $G$. Fig 10 shows the architecture.

**Usage for Targeted Interventions**    In our experiments, we also evaluate the usefulness of causal discovery process to set a prior for the targeted interventions described in Sec 4.2. In Algorithm 1 and 2, we implement this by initializing $p(G)$ to the discovered distribution (rather than uniform).

## E    Additional Results: Diagnosing Causal Misidentification

In Fig 11 we show the causal misidentification in several environments. We observe that while training and validation losses for behavior cloning are frequently near-zero for both the original and confounded policy, the confounded policy consistently yields significantly lower reward when deployed in the environment. This confirms the causal misidentification problem.

## F    DAgger with many more interventions

In the main paper, we showed that DAgger performed poorly with equl number of expert interventions as our method. How many more samples does it need to do well?

The results in Fig 12 show that DAgger requires hundreds of samples before reaching rewards comparable to the rewards achieved by a non-DAgger imitator trained on the original state.

## G    GAIL Training Curves

In Figure 13 we show the average training curves of GAIL on the original and confounded state. Error bars are 2 standard errors of the mean. The confounded and original training curve do not differ significantly, indicating that causal confusion is not an issue with GAIL. However, training requires many interactions with the environment.

Figure 11: An expanded version of Fig 4 in the main paper, demonstrating diagnosis of the causal misidentification problem in three settings. Here, the final reward, shown in Fig 4 is shown in the third column. Additionally, we also show the behavior cloning training loss (first column) and validation loss (second column) on trajectories generated by the expert. The x-axis for all plots is the number of training examples used to train the behavior cloning policy.

## H    Intervention Posterior Inference as Reinforcement Learning

Given a method of evaluating the likelihood $p(\mathcal{O}|G)$ of a certain graph $G$ to be optimal and a prior $p_0(G)$, we wish to infer the posterior $p(G|\mathcal{O})$. The number of graphs is finite, so we can compute this posterior exactly. However, there may be very many graphs, so that impractically many likelihood evaluations are necessary. Only noisy samples from the likelihood can be obtained, as in the case of intervention through policy execution, where the reward is noisy, this problem is exacerbated.

If on the other hand, a certain structure on the policy is assumed, the sample efficiently can be drastically improved, even though policy can no longer be exactly inferred. This can be done in

(a) MountainCar            (b) Hopper

Figure 12: DAgger results trained on the confounded state.

Figure 13: Rewards during GAIL training.

the framework of Variational Inference. For a certain variational family, we wish to find, for some temperature $\tau$:

$$\pi(G) = \underset{\pi(G)}{\arg\min}\, D_{KL}(\pi(G)||p(\mathcal{O}|G)) \tag{4}$$

$$= \underset{\pi(G)}{\arg\min}\, \mathbb{E}_\pi\left[\log p(\mathcal{O}|G) + \log p_0(G)\right] + \tau\mathcal{H}_\pi(G) \tag{5}$$

The variational family we assume is the family of independent distributions:

$$\pi(G) = \prod_i \pi_i(G_i) = \prod_i \text{Bernoulli}(G_i|\sigma(w_i/\tau)) \tag{6}$$

Eq 5 can be interpreted as a 1 step entropy-regularized MDP with reward $\tilde{r} = \log p(\mathcal{O}|G) + \log p_0(G)$ [27]. It can be optimized through a policy gradient, but this would require many likelihood evaluations. More efficient is to use a value based method. The independence assumption translates in a linear Q function: $Q(G) = \langle w, G \rangle + b$, which can be simply learned by linear regression on off-policy pairs $(G, \tilde{r})$. In Soft Q-Learning [16] it is shown that the policy that maximizes Eq 5 is $\pi(G) \propto \exp Q(G)/\tau$, which can be shown to coincide in our case with Eq 6:

$$\pi(G) = \frac{\exp(\langle w, G\rangle + b)/\tau}{\sum_{G'}\exp(\langle w, G'\rangle + b)/\tau} \propto \prod_i \exp(w_i G_i/\tau)$$

$$\implies \pi(G) = \prod_i \frac{\exp(w_i G_i/\tau)}{1 + \exp w_i/\tau} = \prod_i \text{Bernoulli}(G_i|\sigma(w_i/\tau))$$

## Footnotes

[2]We drop time $t$ from the superscript when discussing states and actions from the same time.

[3]More generally, faithfulness places strong constraints on the expert graph. For example, a visual state may contain unchanging elements such as the car frame in Fig 1, which are by definition deterministic functions of the past. As another example, goal-conditioned tasks must include a constant goal in the state variable at each time, which once again has deterministic transitions, violating faithfulness.