[Reviews · NeurIPS 2019]

Reviewer 1



Summary: This paper has a very interesting claim: distributional shift in imitation learning settings is primarily caused by causal misidentification of the features by the learning algorithm. An interesting example is that of a self-driving car policy trained on a dataset of paired image-control datapoints collected by an expert human driving the car. If the images contain the turn signal on the dashboard then the supervised learner learns to have very good predictive power by indexing on that feature in the image. Clearly that does not generalize during test time. While this is a pathological example, such behavior is present in most settings where usually the state is blown-up by appending past states and actions. The authors clearly show this phenomena in various settings like mountain-car, hopper and some selected Atari games. Two methods are proposed as fixes: Interventional setting: In this setting a mixture policy (trained on a uniform distribution of binary masks over the features and an initial seed dataset of states-expert action pairs) is rolled out and the expert is invoked on a subset of encountered states that have the most disagreement with individual binary masks over the features. Note each binary mask encodes a particular causal graph structure. The loss wrt the expert on this subset of states is recorded and a linear regressor trained to result in an exponential distribution over all graphs. Finally the best graph in this distribution is returned. Policy execution setting: In the setting where there expert intervention is not possible, it is assumed that access to episodic rewards is at least present and that such rewards are a good proxy for expert behavior. In this setting the graph parameterized policies are rolled out and the probability of that particular graph is proportionally increased or decreased based on the corresponding episode return. Finally the best graph in this distribution is returned. For image observations as in Atari games a latent space observation via VAE is used to get disentangled observations. Experiments on MountainCar, Hopper, and selected Atari games like Pong, Enduro and UpNDown show that using the methods proposed, results in achieving reward that is similar to that can be achieved using the 'original' state space features (without the confounding previous actions added on) even if the state space is blown-up ('confounded') by appending previous actions. Overall comments: - Pretty well-written paper overall though I have clarification questions below. Overall I find this to be quite an important paper in learning-from-demonstration work and something that will spark quite the discussion at the conference. - Algorithm 1 \pi_{\textrm{mix}} should be cross-referenced to lines 201-202 and ideally introduced there. I am a bit confused between \pi_G and \pi_{\textrm{mix}}. \pi_G is well defined and clear. What precisely is \pi_{\textrm{mix}}? Once I have trained f_{\phi} which according to lines 201-202 is the mixture policy how do I get \pi_{\textrm{mix}}? Note f_{\phi} has a particular signature which takes the appended graph binary vector G. Is it all ones for \pi_{\textrm{mix}}? - What is the disagreement really capturing in Algorithm 1? After walking through some toy examples myself my intuition says that states with high disagreement score by definition have lots of different answers depending on which G is compared against. G's which have causal misidentification will have different answers against the rest of the Gs. Does this assume that causally misidentified Gs are the majority of set of Gs and good Gs are one or few? If so do is it possible for all the bad Gs to agree with each other thus achieving high agreement score? Some light on this will be useful. - How should one set N compared to |G| in Algorithm 1? How many G's were needed in practice in the domains experimented with here? - Table 2: No disc-intervention? - Ultimately from all the experiments is the takeaway that one should not do IL on confounded space (blown-up state space with previous actions appended) in the first place? Is this problem mostly because people tack on the previous actions and exacerbate the causal misidentification issue? Update: ----------- After reading author response I am upgrading my score. An important paper in LfD literature.

Reviewer 2



The work discusses how mis-identifying causal relations can create issues with imitation learning. They show experimental data to support the claim, propose new algorithm to overcome the problem and propose new benchmarks to test the problem and solution. I think this is a very important work that can open a new directions for imitation learning and should be accepted. Detailed remarks: 1) I am not that familiar with causal inference so I am not sure if the algorithm proposed is a good way to identify causal structure, but it makes sense to me and seems to work well empirically. 2) Regarding the GAIL baseline - I think one can explain part of the reason GAIL is successful using causal reasoning (if you get the wrong cause the state space distribution would be different and "caught" by the adversary) and discussed in the paper. 3) In Fig. 11 I would like to see the intervation curve continued to be sure it doesn't converge to a lower state in the end, even if it is higher sooner.

Reviewer 3



This work identifies the problem of causal misidentification in imitation learning and proposes a solution to address this. Current techniques for imitation learning involve learning policies from expert demonstrations without any notion of causality. By simply using behavior cloning, an agent can learn to pay attention to nuisance variables that are not truly important in the real world. If the agent can learn the causal model that affects expert demonstrations, the agent can perform well even when the testing distribution differs from the training distribution. The authors introduce a solution that discovers the correct causal model through intelligent interventions. The approach is evaluated on several domains in which information about past actions is added to the state. Methods that do not learn the causal model wrongly attribute the effect of expert actions to these nuisance correlated factors, while the proposed approach is able to handle this and learns the correct causal model. Strengths: - Causal misidentification is a very relevant and novel problem as current learning methods often fail due to distributional shift. - The proposed approach is an interesting way to capture the relevant factors affecting expert actions. - The clarity of the work is strong. - The examples included in Section 3 set up the problem really well for evaluation experiments later on. Weaknesses: - While the types of interventions included in the paper are reasonable computationally, it would be important to think about whether they are practical and safe for querying in the real world. - The assumption of disentangled factors seems to be a strong one given factors are often dependent in the real world. The authors do include a way to disentangle observations though, which helps to address this limitation. Originality: The problem of causal misidentification is novel and interesting. First, identifying this phenomenon as an issue in imitation learning settings is an important step towards improved robustness in learned policies. Second, the authors provide a convincing solution as one way to address distributional shift by discovering the causal model underlying expert action behaviors. Quality: The quality of the work is high. Many details are not included in the main paper, but the appendices help to clarify some of the confusion. The authors evaluated the approach on multiple domains with several baselines. It was particularly helpful to see the motivating domains early on with an explanation of how the problem exists in these domains. This motivated the solution and experiments at the end. Clarity: The work was very well-written, but many parts of the paper relied on pointers to the appendices so it was necessary to go through them to understand the full details. There was a typo on page 3: Z_t → Z^t. Significance: The problem and approach can be of significant value to the community. Many current learning systems fail to identify important features relevant for a task due to limited data and due to the training environment not matching the real world. Since there will almost always be a gap between training and testing, developing approaches that learn the correct causal relationships between variables can be an important step towards building more robust models. Other comments: - What if the factors in the state are assumed to be disentangled but are not? What will the approach do/in what cases will it fail? - It seems unrealistic to query for expert actions at arbitrary states. One reason is because states might be dangerous, as the authors point out. But even if states are not dangerous, parachuting to a particular state would be hard practically. The expert could instead be simply presented a state and asked what they would do hypothetically (assuming the state representations of the imitator and expert match, which may not hold), but it could be challenging for an expert to hypothesize what he or she would do in this scenario. Basically, querying out of context can be challenging with real users. - In the policy execution mode, is it safe to execute the imitator’s learned policy in the real world? The expert may be capable of acting safely in the world, but given that the imitator is a learning agent, deploying the agent and accumulating rewards in the real world can be unsafe. - On page 7, there is a reference to equation 3, which doesn’t appear in the main submission, only in the appendix. - In the results section for intervention by policy execution, the authors indicate that the current model is updated after each episode. How long does this update take? - For the Atari game experiments, how is the number of disentangled factors chosen to be 30? In general, this might be hard to specify for an arbitrary domain. - Why is the performance for DAgger in Figure 7 evaluated at fewer intervals? The line is much sharper than the intervention performance curve. - The authors indicate that GAIL outperforms the expert query approach but that the number of episodes required are an order of magnitude higher. Is there a reason the authors did not plot a more equivalent baseline to show a fair comparison? - Why is the variance on Hopper so large? - On page 8, the authors state that the choice of the approach for learning the mixture of policies doesn’t matter, but disc-intervention obtains clearly much higher reward than unif-intervention in Figures 6 and 7, so it seems like it does make a difference. ----------------------------- I read the author response and was happy with the answers. I especially appreciate the experiment on testing the assumption of disentanglement. It would be interesting to think about how the approach can be modified in the future to handle these settings. Overall, the work is of high quality and is relevant and valuable for the community.

[Author Response · NeurIPS 2019]

We would like to thank all reviewers for taking the time to review our work and for providing thoughtful suggestions.

**R1: Avoid misattribution by using "original" space?** Real-world problems often have no obvious or easily engi-
neered "original space", as the success of deep feature learning attests. Very natural-seeming state spaces may still
include nuisance factors—as in our dashboard light setting (Fig 1), where causal misattribution occurs when using the
full image from the camera (scenario A). Our testbeds (Sec 3.2) do indeed introduce more deliberate nuisance variables
for ease of evaluation, but evidence suggests that misattribution is pervasive in common imitation learning settings.
For example, history would seem a natural part of the original state space for real-world driving, yet as shown in Pg 4,
recurrent/history-based imitation has been observed repeatedly in prior work to hurt performance.

**R3: better intervention methods?** Policy execution and expert queries are intervention modes that are close to
reinforcement learning and DAgger-style behavior cloning respectively, and inherit their strengths and weaknesses.
Specific settings might indeed warrant other types of interventions for safety/practicality. As an example, the learner
could solicit preferences (see Wirth et al, "A survey ...", 2017) that indicate that the interventional episodes of some
hypotheses $G$ are to be preferred over others, so that the expert need not be parachuted or placed in dangerous states. In
all cases however, an intervention *must* involve some action by a suboptimal policy in the environment, which naturally
incurs some risk—after all, causal misidentification can only be detected under distributional shift from the demos.

**R3: What if state isn't disentangled?** Then, individual dimensions in the state might capture both causes as well as
nuisance variables. The problem of discovering true causes is no longer reducible to searching over $2^m$ graphs. To test
this empirically, we create a variant of our MountainCar testbed, where the 3-D past action-augmented state vector is
rotated by a fixed, random rotation. After training the graph-conditioned policies and applying 30 episodes of policy
execution intervention or 20 expert queries, we get -145 and -165 reward respectively. This is significantly lower than
in the disentangled (non-rotated) setting, indicating disentanglement is important for the effectiveness of our approach.

**R1: Why disagreement?** Choosing new samples to label based on disagreement among a committee (Seung et al
1992, "Query by Committee") is a widely used heuristic in active learning, with this simple intuition: to disambiguate
among competing hypotheses $G$, the most informative states are those that induce maximum disagreement among
the hypotheses. Our empirical results, in agreement with many prior active learning approaches, suggests that this
heuristic works well in practice. Still, it is an imperfect heuristic, and it is certainly possible to construct cases, as
R1 suggests, where useful states may be discarded because of high agreement among bad hypotheses. However, Alg
1 works because it need not identify every single useful expert query intervention — it suffices to identify a small
subset of good interventions. It might be possible to do better than Alg 1 by updating the distribution over graphs after
each query and recomputing the disagreement based on this rebalanced distribution. This would iteratively bias the
distribution towards better-performing graphs, so that for later queries, the disagreement score would more effectively
measure mismatch between good and bad graphs. We will experiment with this for future versions.

'**R1: What exactly is $\pi_{mix}$ in Alg 1?**' $\pi_{mix}$ samples a random graph $G$ per episode and then executes $\pi_G$, with $G$
concatenated to the state. We will clarify this. **R1: how many G's?** $|G| = 2^m$, with $m$ being $2 + 1$ (state+action)
dimensions) for MountainCar, $11 + 3$ for Hopper and 30 for Atari (dimensionality of VAE latent). We observe that the
number of interventions required, $N$, increases as $|G|$ increases (see Fig 7 + Ln323-339). **R1: DISC-INTERVENTION in
Tab 2?** DISC-INTERVENTION, which employs a variational approach to causal discovery, gets progressively harder to
train as the state space increases. Already on 14-D Hopper states, it does no better than UNIF-INTERVENTION. On Atari
envs in Tab 2, DISC-INTERVENTION ran into optimization difficulties and yielded poor results. We will add a note.
**R3: Typos & appendices.** We will fix the typo $Z_t \rightarrow Z^t$. The reference to Eq 3 in page 7 should be Eq 1, we will
fix this. We will attempt to move more appendices into the main paper, and we will release source code to remove
any ambiguity. **R3: update time cost?** Yes, the model is updated after each episode, but this is very fast online
linear regression, taking negligible time compared to executing an episode from the neural net policy. **R3: size of
disentangled state space?** For Atari, we set the VAE latent size heuristically, to be as small as possible, but still
produce good reconstructions, as assessed visually. **R3: DAgger sparse evaluation?** As shown in Fig 11, DAgger
takes 20x for MountainCar and 600x for Hopper more queries to match the performance of our method — to keep
the number of experiments and computation costs manageable, its performance is evaluated more sparsely. **R3: High
variance in Hopper?** This is down to Hopper being inherently unstable, where some random seeds for all methods
result in the Hopper falling over, producing very poor rewards. **R3: More equivalent baselines than GAIL?** GAIL
is a strong baseline for imitation, and we do not know of other methods that would be more equivalent/competitive
with our approach in these settings. **R3: DISC-INTERVENTION good?** DISC-INTERVENTION does indeed perform well on
low-dimensional state spaces ( e.g. MountainCar), but runs into optimization difficulties as the state space grows. Also
see our response to R1 above.
**R2: Does performance degrade with more interventions?** In all our experiments, performance was approximately
monotonic: with more data from interventions, performance either stabilizes or significantly improves. We will evaluate
UNIF-INTERVENTION at more expert queries in future revisions for a more complete version of Appendix Fig 11.

[Meta-Review · NeurIPS 2019]

The work discusses how mis-identifying causal relations can create issues with imitation learning. They show experimental data to support the claim, propose new algorithm to overcome the problem and propose new benchmarks to test the problem and solution. All reviewers think very highly of the paper, and suggested acceptance.